# Detection of *Trypanosoma cruzi* in a Reactive Blood Bank Sample in Sonora, Mexico

**DOI:** 10.3390/tropicalmed10040104

**Published:** 2025-04-11

**Authors:** Idalia Paredes-Sotelo, Mónica Reséndiz-Sandoval, Adriana Garibay-Escobar, Edgar Alfonso Paredes-González, Aracely Angulo-Molina, Angel Ramos-Ligonio, Eric Dumonteil, Claudia Herrera, Olivia Valenzuela

**Affiliations:** 1Departamento de Ciencias Químico Biológicas, Universidad de Sonora, Hermosillo C.P. 83000, Sonora, Mexico; a214200158@unison.mx (I.P.-S.); mresendiz@ciad.mx (M.R.-S.); adriana.garibay@unison.mx (A.G.-E.); aracely.angulo@unison.mx (A.A.-M.); 2Laboratorio de Inmunología, Centro de Investigación en Alimentación y Desarrollo A. C., Hermosillo C.P. 83304, Sonora, Mexico; 3Departamento de Ciencias Químico Biológicas y Agropecuarias, Universidad de Sonora, Caborca C.P. 83600, Sonora, Mexico; edgar.paredes@unison.mx; 4LADISER, Inmunología y Biología Molecular, Facultad de Ciencias Químicas, Universidad Veracruzana, Orizaba C.P. 94340, Veracruz, Mexico; angramos@uv.mx; 5Asociación Chagas con Ciencia y Conocimiento A. C., Orizaba C.P. 94390, Veracruz, Mexico; 6Department of Tropical Medicine and Infectious Disease, Celia Scott Weatherhead School of Public Health and Tropical Medicine, Tulane University, New Orleans, LA 70112, USA; edumonte@tulane.edu (E.D.); cherrera@tulane.edu (C.H.)

**Keywords:** *Trypanosoma cruzi*, Chagas disease, blood bank, serology, molecular diagnostics, discrete typing unit

## Abstract

Chagas disease is a neglected disease caused by the parasite *Trypanosoma cruzi*, a public health problem in both endemic and non-endemic countries. In Mexico, the southern region is considered endemic, and cases are frequently reported; however, in the northwestern region, only a few cases are confirmed annually. This study describes, for the first time, the Discrete Typing Unit (DTU) of *Trypanosoma cruzi* in a volunteer blood donor rejected for being reactive in the northwestern region of Mexico. Seroreactivity was confirmed using “in-house” ELISAs which employed three different antigens: total extract from *Trypanosoma cruzi* isolated from a vector (*Triatoma rubida*) from Sonora (strain T1), strain H1 and CL-Brener. The molecular characterization of *Trypanosoma cruzi* was conducted by amplifying satellite DNA by qPCR and posterior sequencing of the mini-exon gene, using Next Generation Sequencing (NGS) to enhance the accuracy of genetic characterization. The results show that the reactive status of this blood donor was confirmed using our in-house ELISAs, and the presence of *Trypanosoma cruzi* by detecting TcI DTU confirmed the infection status.

## 1. Introduction

Chagas Disease (CD) is one of the neglected tropical diseases of greatest public health importance in the Americas, caused by *Trypanosoma cruzi* (*T. cruzi*). About 6–7 million people worldwide, mainly in Latin America, are estimated to be infected with *T. cruzi* [1]. The disease has two phases, acute and chronic, in which patients may transition from an acute phase to a chronic infection. Usually, the acute phase lasts from 4 to 8 weeks, and is characterized by high parasitemia and mild symptoms which can be resolved spontaneously [2]. In the chronic phase, most infected people remain asymptomatic; however, 30–40% will develop cardiac and/or digestive or neurological involvement within 10–30 years [3]. The persistence of Chagas disease is linked to social, cultural, historical, political, and economic processes [4].

*T. cruzi* exhibits great genetic variability and is classified into seven Discrete Typing Units (DTUs) from TcI-TcVI and Tc-Bat [5,6]. The term is defined as “sets of stock that are genetically more related to each other than to any other stock and that are identifiable by common genetic, molecular or immunological markers” [7]. This genetic diversity of *T. cruzi* impacts its biological characteristics, such as antigenic profile, virulence factors, or variability in the efficacy of the only two drugs available for CD, Benznidazole (BNZ) and Nifurtimox (NFX) [8,9,10].

Chagas disease was once delimited to vulnerable rural areas where the vectors reside and multiple reservoir hosts of *T. cruzi* parasites, including sylvatic and domestic animals throughout the Americas, meaning that the infection could not be eradicated [11]; however, due to human dynamics, the epidemiological patterns have changed with urbanization. Triatomine bugs, known as kissing bugs, are the main vectors. Until now, 157 species have been classified into five tribes and 18 genera [12,13]. In Northwestern Mexico, six triatomine species have been reported: *Triatoma protacta*, *T. incrassata*, *T. recurva*, *T. rubida*, *T. sinaloensis*, and *Paratriatoma hirsuta* [14,15,16,17].

Although triatomines are found widespread across Latin America, in some regions, the population remains unaware of the risk associated with contact. These hematophagous bugs emerge at night to seek a blood meal, making it difficult for individuals to know whether they have been bitten or exposed by living in or visiting infested areas [18]. The bug defecates near the bite site, and the parasite’s infective stage can enter the body when the host rubs or scratches, allowing the passage of *T. cruzi* through breaks in the skin or conjunctiva; however, there are other forms of transmission, such as blood transfusion, congenital infection, organ transplantation, and oral transmission [3].

Blood transfusion is an important route of transmission of *T. cruzi*. Although blood bank screening has been regulated in Mexico since 2012, according to the Official Mexican Standard (NOM 253-SSA2-2012), few confirmed cases have been reported, even though infected triatomines have been reported in many regions of the country [19]. Today, screening for Chagas is only carried out on blood donors, which leaves most of the population unaware of whether they are infected and, if they are infected, without access to treatment [20]. The WHO recommends screening blood donors as well as organ and tissue donors in endemic areas, early access to diagnosis, treatment and follow-up, and screening of newborns and other children of infected mothers, to increase detection and care of the affected population all over the world [11]. Chagas disease is endemic in most of the countries of America, including Mexico. However, this disease is poorly known in the Northwest region, probably due to the low cases reported annually. This situation could explain the limited strategies to control, prevent, and monitor epidemiological surveillance.

In 2023, Mexico reported 99 acute cases (0.08/100,000) and 891 chronic cases of Chagas (0.69/100,000). The states of Veracruz and Yucatan, located in the southern region of Mexico, showed the highest incidence of chronic cases of Chagas (2.54 and 2.44/100,000, respectively). In Sonora, a predominantly arid region and the second-largest state of Mexico, only three chronic cases have been reported, with an incidence of 0.09/100,000 [21]. In contrast, Chagas disease is more prevalent in southwestern Mexico, where the climate ranges from temperate to tropical.

There is currently a wide variety of serological tests for the diagnosis of *T. cruzi* infection, including commercial tests with high sensitivity and specificity (>98%), which are mainly based on parasite antigens (recombinants or total extracts); however, no gold standard has been identified for an accurate and reliable diagnostic of *T. cruzi* infection, which is a limiting factor for these tests. The World Health Organization and most National guidelines in Latin American countries still recommend the use of two tests based on different principles and antigens, and in case of discordance, a third test can provide a final diagnosis [22].

Polymerase Chain Reaction (PCR) and Real-Time quantitative PCR (qPCR) offer high sensitivity for detecting DNA, and appear as opportunities to monitor parasitic levels in the bloodstream, and several studies have evaluated their high sensitivity and specificity in amplifying nuclear DNA (SAT-DNA) and kinetoplast DNA (kDNA) sequences [23,24]. There is not a qualitative or quantitative standardized/validated PCR for *T. cruzi* DNA detection in diagnostic routine [23]. So far, there is no gold standard qPCR protocol to diagnose *T. cruzi* infections; however, the use of molecular tools supports the accuracy of these diagnostic tools [24].

The objective of this study was to analyze the presence of anti-*T. cruzi* antibodies using “in-house” ELISAs which employed as antigen a total extract from *Trypanosoma cruzi* isolated from triatomine vectors from Sonora, strain H1 and CL-Brener, respectively, and perform the molecular *T. cruzi* characterization in a reactive sample of a volunteer blood donor screened at the Sonora Mexico blood bank.

## 2. Materials and Methods

A 38-year-old male hemodonor was identified by the blood bank of Sonora, Mexico, as a reactive case of *T. cruzi*. Two blood samples were taken (in the absence and presence of EDTA-anticoagulant).

The presence of anti-*T. cruzi* antibodies were analyzed by “in-house” ELISAs, which are customized assays using three different parasite extracts: (a) TRUB/MEX/2022/T1/TcI isolated was obtained from an insect triatomine (*Triatoma rubida*) from Sonora, Mexico; (b) MHOM/MEX/0000/H1/TcI isolated from a patient from Yucatan, Mexico; and (c) TINF/BR/1963/CL-Brener/TcVI a reference strain, modified from Guzmán-Gómez et al., (2015) [22]. Briefly, 96-well clear flat bottom microplates (Corning, Inc., New York, NY, USA) were coated with 5 µg/mL of each crude antigen extract overnight at 4 °C and blocked with Bovine Serum Albumin (BSA) (Jackson Immuno Research, Baltimore, PA, USA) 2% for 1 h at room temperature. 

The serum sample was tested individually against each antigen in ELISAs. These protocols ensure both technical repeatability (triplicate wells) and biological repeatability (sample repetition on different days). The serum was diluted 1:100 in non-fat milk 1% in Phosphate-Buffered Saline (PBS)—tween 0.05% and incubated for 30 min at room temperature. After washing, goat anti-human IgG-HRP (Horseradish Peroxidase) (Sigma, St. Louis, MO, USA) was incubated for 30 min at room temperature. After a final wash, 3,3′,5,5′-tetramethylbenzidine (TMB) substrate solution (Immunochemistry, Hennepin, MN, USA.) was added and incubated at room temperature, protected from the light, and the reaction was stopped after 15 min with 1 M sulfuric acid (H_2_SO_4_) (Sigma-Aldrich, St. Louis, MO, USA). Absorbance was read on a microplate absorbance reader at 450 nm (Thermo Scan Thermofisher, Waltham, MA, USA). The cut-off OD (optical density) value for commercial ELISA tests was determined according to each manufacturer’s recommendations. For in-house ELISA tests, the cutoff point was defined as the mean OD of negative serum samples (n = 20) plus three standard deviations. In order to be able to compare the results of the different ELISA assays used, the results are presented as the resulting value of the optical density/cut-off line of each assay [25]. The Chagas III ELISA diagnostic kit (BiosChile, Santiago, Chile) was also used according to the supplier’s instructions.

Genomic DNA was extracted from an EDTA-blood sample using the DNeasy Blood and Tissue kit (QIAGEN, Valencia, CA, USA) following the manufacturer’s instructions. Real-time PCR (qPCR), also known as quantitative PCR, was used to amplify the satellite DNA of *T. cruzi* using StepOne Real-Time PCR Systems (Applied Biosystems™, Thermo Fisher Scientific, Waltham, MA, USA). The primers used for amplification (TCZ3-Fw: TGC TGC AST CGG CTG ATC GTT TTC GA and TCZ2-Rv: CCT CCA AGC AGC GGA TAG TTC AGG) were described previously [26], using Brillant III ultra-fast SYBR green PCR (Agilent Technologies, Santa Clara, CA, USA). The PCR cycling conditions consisted of an initial step of 3 min at 95 °C followed by 40 cycles of denaturalization at 95 °C for 10 s and annealing/extension at 60 °C for 10 s. The sample was considered positive for *T. cruzi* if the Cycle Threshold (Ct) was <40. Each reaction included positive (DNA from a *T. cruzi*-positive sample) and negative controls (nuclease-free water). Amplification specificity was confirmed by melting curve analysis, with a single peak indicating specific product amplification.

As previously reported, discrimination between the discrete typing units (DTUs) was performed using a Nested-PCR for the amplification of the mini-exon gene with the primers UTCC, TC, and TC2 [27]. The oligonucleotide sequences of the primers are UTCC-Fw: CGT ACC AAT ATA GTA CAG AAA CTG, TC2-Rv: CCT GCA GGC ACA CGT GTG TG and TC-Fw: CCC CCC TCC CAG GCC ACA CTG. For the primary PCR, the DNA sample was used as the template, UTCC-Fw and TC2-Rv were used for primers. For the secondary PCR, 1 µL of primary PCR product was used as the template, the primers were TC-Fw and TC2-Rv. Each PCR consisted of 30 cycles of denaturation at 94 °C for 30 s, annealing at 55 °C for 30 s, and final extension at 72 °C for 30 s, with an initial denaturation of 5 min at 94 °C and a final extension at 72 °C for 5 min. The PCR products were separated by electrophoresis (2% agarose). The expected product size is 300 base pairs for TcII-TcVI or 350 base pairs for TcI. Amplicon was purified for sequencing on MiSeq (Illumina San Diego, CA, USA) platform. Sequence files were imported in Geneious 11.0.2 software for analysis and mapped using *T. cruzi* reference sequences as previously described [28]. Phylogenetic analysis was performed using the approximately maximum likelihood method implemented in FastTree with 1000 bootstrap replicates to confirm DTU classification and relationship with reference sequences.

## 3. Results and Discussion

Although 65 years have passed since the first case of CD was reported in Guaymas, Sonora [29], this study describes, for the first time, the presence of *T. cruzi* in a volunteer blood donor from the northwestern region of Mexico. The blood donor was 38 years old, worked in mining, and lived in a house with a cement floor and thatched roof. He reported having pets at home but did not recognize nor know triatomine vectors. In 2015, he visited a rural area and, at the age of 17, visited Tucson, Arizona. He had never received a blood transfusion or donated organs, and donated blood in 2008 without *T. cruzi* screening. In 2018, he donated blood again and was found reactive for *T. cruzi* by the blood bank. Remarkably, there was no subsequent follow-up or monitoring to assess clinical status, potential treatment or further epidemiological surveillance. This lack of follow-up increases the risk of disease progression and is a missed opportunity for early treatment.

The treatment of Chagas disease has been ongoing for 50 years and relies on two drugs: BNZ and NFX. These have great effectiveness in acute infections but significant side effects due to the oxidative stress reactive metabolites generated, which can eliminate *T. cruzi*, but harm the organism [2]. It is not clear whether these drugs can effectively cure patients infected with other *Trypanosoma* species due to metabolic differences, but they have been tested in studies as a combination of drugs [30,31].

Despite the high discordance between the serologic methods used for the diagnosis of Chagas in Mexico [22], the analysis of the serum sample showed the presence of anti-*T. cruzi* IgG antibodies using the three “in-house” ELISA tests, which employed each extract antigens (T1, H1 and CL Brener) independently, observing a greater reactivity to the extracts of the T1 and H1 isolates (TcI DTU) compared with the CL-Brener reference strain (TcVI) (Figure 1). In addition, the sample was also reactive using the commercial diagnostic kit Chagas III ELISA diagnostic kit (BiosChile, Santiago, Chile).

It has been stated that the genetic variability of the parasite may affect the performance of the serological tests, due to the different antigenic profiles. Hence, previous studies have recommended employing antigen extracts from local parasites to enhance the sensitivity of the test [32]. Regarding actual serological diagnostic methods, the use of total antigenic extracts of *T. cruzi* has shown good results compared to commercial tests [4]. The present results clearly distinguish that the use of the local strain allowed the recognition of antibodies specific to *T. cruzi* whole antigens obtained from the vector *Triatoma rubida* of Sonora, México (local strain).

It has also been observed that the detection of the parasite’s genetic material using molecular tools is a very important diagnostic complement, despite its limitations, such as low or intermittent parasitemia in the chronic phase [24,33]. The molecular detection of *T. cruzi* DNA offers multiple advantages that can enhance the reliability of the molecular techniques [24].

The presence of *T. cruzi* in the blood sample was confirmed by qPCR (Ct = 33), and the mini-exon gene was amplified, producing the characteristic band of TcI DTU (350 bp) (Figure 2), which was confirmed by sequencing (GenBank Accession Number PP993262) and phylogenetic analysis (Figure 3), indicating that it is closely related to a TcI sequence from a patient from Yucatan, Mexico. The detection of a TcI DTU is not surprising since it is widely distributed in the Mexican territory [34]; however, most isolates come from vectors. In addition, this DTU is not only associated with sylvatic cycles, but also with domestic cycles and clinical manifestations such as Chagasic cardiomyopathy [35]. The detection of *T. cruzi* in the blood donor sample is alarming, because it may imply an active risk for transfusion-related transmissions, as there are discordant results between serological methods currently employed for CD screening. This study confirmed the presence of the *Trypanosoma cruzi* by qPCR in a volunteer blood donor rejected for being reactive in the northwestern region of Mexico.

The risk of *T. cruzi* infection after a transfusion from an infected donor ranges between 10 and 25% depending on the blood component and increases in polytransfused patients [18]. Data from the National Blood Transfusion Center in Mexico show that, in the state of Sonora, 163,416 transfusions were recorded from 2018 to 2020. This suggests that, without proper screening, more than 160,000 people could be at risk of acquiring the infection. For several decades, total extracts of the parasite have been used for diagnosis and shown to provide good results [36]. However, due to the diversity of the parasite, the use of antigens from local strains has been recommended to ensure diagnosis, as they present a greater number of antigens that may be recognized by antibodies [22].

## 4. Conclusions

The serological analysis demonstrated that the “in-house” ELISA, using total parasite extracts from local strains, proved effective in detecting anti-*T. cruzi* antibodies. Additionally, molecular techniques confirmed the presence of TcI DTU in the blood sample, further supporting the accuracy of these diagnostic tools. The presence of vectors infected with *T. cruzi* suggests that vectorial transmission in the state of Sonora is an underexplored area. This highlights the urgent need to implement vector control programs and epidemiological monitoring of Chagas disease, which would allow for timely detection and early treatment.

Considering the risk associated with transfusion-transmitted infections, it is crucial to ensure rigorous testing of blood donors and enhance public health strategies to mitigate the spread of Chagas disease.

## Figures and Tables

**Figure 1 tropicalmed-10-00104-f001:**
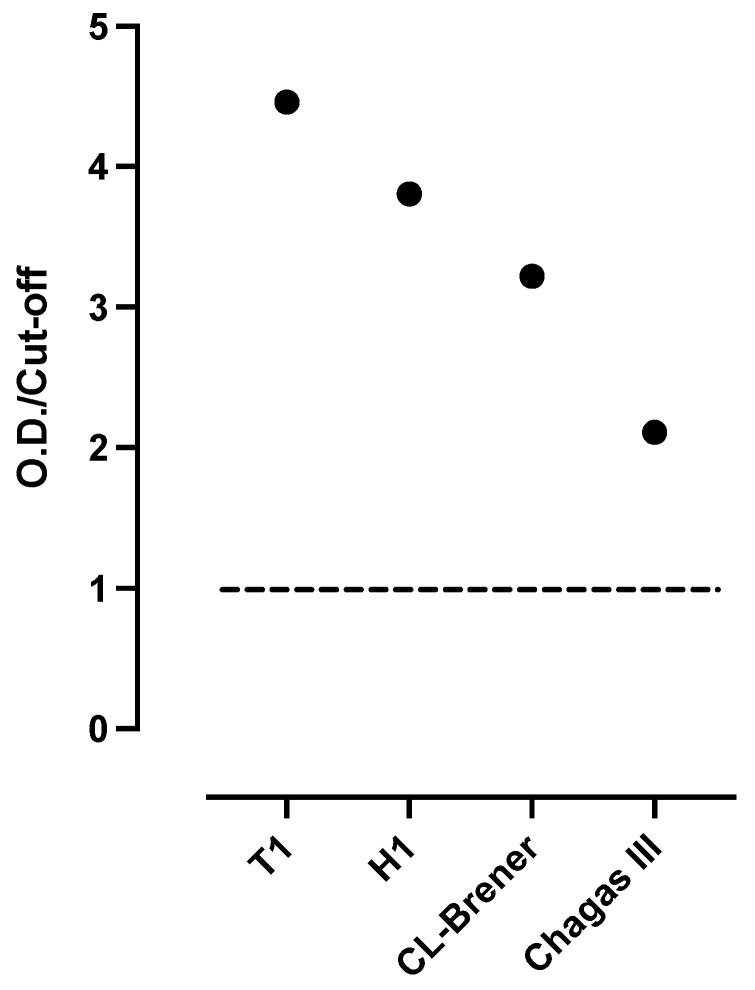
Presence of anti-*Trypanosoma cruzi* IgG antibodies in serum of blood donor. Detection of anti-*T. cruzi* IgG antibodies was performed using four ELISA tests. Three in-house ELISAs, utilizing whole parasite extracts from T1, H1, and CL-Brener strains, respectively, and one commercial kit (Chagas III) were used. Optical densities relative to the cut off (OD/cut off) are shown. The horizontal line of value 1 represents the cut off.

**Figure 2 tropicalmed-10-00104-f002:**
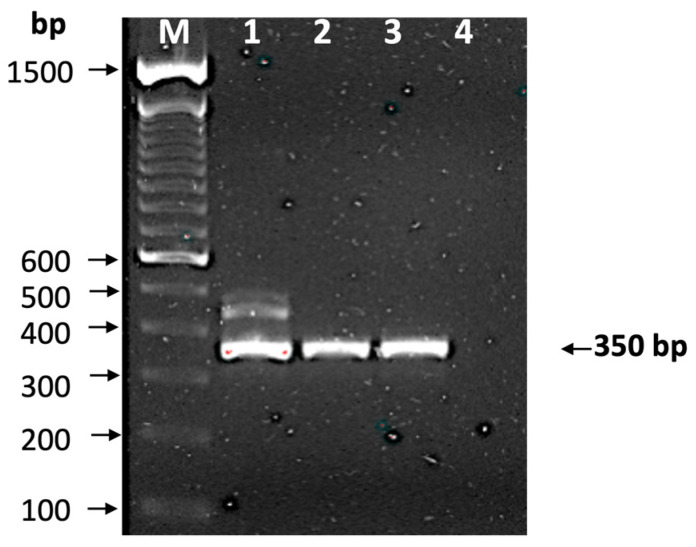
Identification of discrete typing unit. The mini-exon gene was amplified and observed in 2% agarose gel stained with ethidium bromide. Lane M, molecular markers (bp); lane 1, isolate T1; lane 2, isolate H1; lane 3, reactive sample from blood bank; and line 4, negative control. The arrow indicates a 350 bp amplification, which corresponds to TcI.

**Figure 3 tropicalmed-10-00104-f003:**
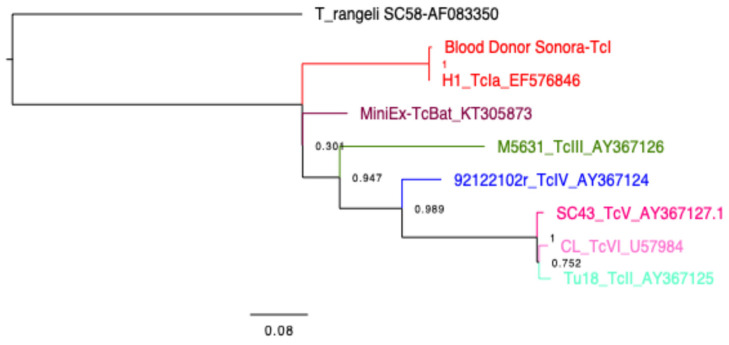
Phylogenetic tree. Constructed with FastTree 2.1.11 and Jukes–Cantor models of nucleotide evolution, to compare the parasite sequence from the blood donor with those from reference sequences (Strains and Genbank accession numbers indicated in the tree). Parasite DTUs are color-coded as indicated and a *T. rangeli* sequence is used as outgroup. The reliability of each split is based on local support values with the Shimodaira–Hasegawa test and resampling the site maximum likelihoods 1000 times.

## Data Availability

The data underlying this article are available in the article, and the sequences are available at GenBank (Accession Number PP993262).

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
