# Peer review of "Detection of Trypanosoma cruzi in a Reactive Blood Bank Sample in Sonora, Mexico"

_tropicalmed, 2025, doi:10.3390/tropicalmed10040104_

Round 1

Reviewer 1 Report

Comments and Suggestions for Authors

I congratulate the authors for this manuscript.

I consider that it is an important piece of information that could be helpful to detect the presence of T. cruzi in blood donors and others. The analysis in additional donors is of course needed.

I only would like to ask  if there is a description at least of the health of the 38 year old potential donor, at the time when the positive reaction was detected.

Author Response

Reviewer 1

  1. I congratulate the authors for this manuscript.

Answer: Thank you for your positive comments

  1. I consider that it is an important piece of information that could be helpful to detect the presence of T. cruzi in blood donors and others. The analysis in additional donors is of course needed.

Answer: Thank you for your valuable comments.

  1. I only would like to ask if there is a description at least of the health of the 38 year old potential donor, at the time when the positive reaction was detected.

Answer:  We appreciate your valuable contribution, which we agree with it. We have compiled all the information available at the blood bank. The donor does not report symptoms related to ChD

Reviewer 2 Report

Comments and Suggestions for Authors

In the title, the word "bank" is mentioned twice. Since it is implicit that the analyzed sample came from a blood bank, I recommend revising the title to: "Detection of Trypanosoma cruzi in a Reactive Blood Bank Sample in Sonora, Mexico."

The use of Next Generation Sequencing (NGS) should be emphasized in the abstract, as this tool offers greater robustness in determining the sample's DTU. Additionally, it is important to specify the bioinformatics software used for genetic data analysis.

The information provided to confirm the infection by serology is unclear. It states that an in house ELISA was performed using the antigenic extracts from the H1 and CL-Brener strains and of T. cruzi vectors. Was the antigen isolated from more than one vector? Has this in house ELISA been executed with a pool of three total antigenic extracts or house one separately? Figure 1 presents the results of four different ELISAs: three utilizing different antigens and one using a commercial kit that was not mentioned in methodology . Furthermore, Figure 1 needs clarification on why there are two axes showing different absorbance values at the same wavelength. There is also no information provided regarding the number of repetitions conducted with the sample or the mean value obtained from the results. Additionally, the explanation of Figure 1 could be enhanced for clarity.

In the methodology section, several acronyms are introduced without explanation, such as BSA, PBS, HRP, and TMB. There is also no mention of the intercalating fluorophore used, was it SYBR Green or EvaGreen? Additionally, the quantity of the sample used in the second nested PCR is not specified.

Finally, there is no information regarding the clinical status of the analyzed patient, including whether they received treatment. While DTU I is typically associated with sylvatic reports and asymptomatic human clinical cases, severe symptomatic cases linked to this DTU, particularly those resulting from oral infections, have also been documented. Also I reccomend to review, the English writing for clarity and precision. 

Author Response

In the title, the word "bank" is mentioned twice. Since it is implicit that the analyzed sample came from a blood bank, I recommend revising the title to: "Detection of Trypanosoma cruzi in a Reactive Blood Bank Sample in Sonora, Mexico."

Answer: The changes were made accordingly.

Page 1, Lines 2-3…..Detection of Trypanosoma cruzi in a reactive blood bank sample in Sonora, Mexico

The use of Next Generation Sequencing (NGS) should be emphasized in the abstract, as this tool offers greater robustness in determining the sample's DTU. Additionally, it is important to specify the bioinformatics software used for genetic data analysis.

Answer: The changes were made accordingly.

Page 1, Lines 31-33: ….. sequencing the mini-exon gene, using Next Generation Sequencing (NGS) to enhance the accuracy of genetic characterization.

The information provided to confirm the infection by serology is unclear. It states that an in house ELISA was performed using the antigenic extracts from the H1 and CL-Brener strains and of T. cruzi vectors. Was the antigen isolated from more than one vector?

Answer: Thank you for your valuable comments. Was isolated from one vector.

Page 3, Lines 119-124… The presence of anti-T. cruzi antibodies were analyzed by “in-house” ELISAs, which are a customized assays using three different parasite extracts: a) TRUB/MEX/2022/T1/TcI isolated was obtained of an insect triatomine (Triatoma rubida) from Sonora, Mexico; b) MHOM/MEX/0000/H1/TcI isolated from a patient from Yucatan, Mexico; and c) TINF/BR/1963/CL-Brener/TcVI a reference strain, modified from Guzmán-Gómez et al., (2015)

Has this in house ELISA been executed with a pool of three total antigenic extracts or house one separately?

Answer: “in-house” ELISAs were used to identify antibodies anti-Trypanosoma cruzi, employing each whole parasite extract separately.

Figure 1 presents the results of four different ELISAs: three utilizing different antigens and one using a commercial kit that was not mentioned in methodology.

Answer: we have included the information accordingly:

Page 4, Lines 141-142: Chagas III ELISA diagnostic kit (BiosChile, Santiago, Chile) was also used according to the supplier's instructions.

Furthermore, Figure 1 needs clarification on why there are two axes showing different absorbance values at the same wavelength.

Answer: we thank you for your feedback and apologize for the oversight regarding the commercial ELISA assay reading wavelength, which was 450 nm. In the revised Figure 1, we now present the data as DO/cut ratios, facilitating direct comparison between samples. Additionally, we have made the clarification in the section of materials and methods

Page 3-4, lines 136-141… The cut-off OD (optical density) value for commercial ELISA tests was determined according to each manufacturer's recommendations. For in-house ELISA tests, the cutoff point was defined as the mean OD of negative serum samples (n = 20) plus three standard deviations. In order to be able to compare the results of the different ELISA assays used, the results are presented as the resulting value of the optical density / cut-off line of each assay [25]

There is also no information provided regarding the number of repetitions conducted with the sample or the mean value obtained from the results.

 Answer: we have included the information accordingly:

Page 3, Lines 126-129 : The serum sample was tested individually against each antigen. These protocols ensure both technical repeatability (triplicate wells) and biological repeatability (sample repetition on different days).

Additionally, the explanation of Figure 1 could be enhanced for clarity.

Answer: the caption was rephrased

In the methodology section, several acronyms are introduced without explanation, such as BSA, PBS, HRP, and TMB.

Answer: Thank you for your suggestion; we have included the information accordingly: Page 3, Line 126: … with Bovin Serum Albumin (BSA) 2% for 1 h at room temperature Page 3, Lines 129-130: … 1% in Phosphate-Buffered Saline (PBS) - tween 0.05%. Page 3, Line 131: … IgG-HRP (Horseradish Peroxidase) (Sigma, St. Louis, MO, USA)…Page 3, Lines 132-133: … wash, 3,3',5,5'-tetramethylbenzidine (TMB) substrate solution

There is also no mention of the intercalating fluorophore used, was it SYBR Green or EvaGreen?

Answer: Thank you for your suggestion; we have included the information accordingly:

Page 4, Line 149: …using Brillant III ultra-fast SYBR green PCR (Agilent Technologies).

Additionally, the quantity of the sample used in the second nested PCR is not specified.

Answer: Thank you for your suggestion, we have included the information.

Page 4, Lines 161-162:… For the secondary PCR, 1 µL of primary PCR product was used as the template,

Finally, there is no information regarding the clinical status of the analyzed patient, including whether they received treatment.

Answer: We agreed that this information is important, so we included the information provided by the blood bank. However, blood bank does not contain any treatment data since the patient did no present any symptoms of disease at the time of donation. Further studies should focus on the epidemiology of CD to understand the most common transmission routes, as well as the evolution during case treatment.

 Although this case report does not provide information on the clinical history and its follow-up, it opens the way for future investigations to evaluate the presence of T. cruzi in this area as well as the dynamics of transmission of the disease. We appreciate your attention and hope that these arguments will enhance the discourse on the importance of our study.

While DTU I is typically associated with sylvatic reports and asymptomatic human clinical cases, severe symptomatic cases linked to this DTU, particularly those resulting from oral infections, have also been documented. Also I reccomend to review, the English writing for clarity and precision. 

Answer: Thank you for your suggestion, we have included the information.

Page 6, Lines 231-232 …. In addition, this DTU is not only associated with sylvatic cycles, but also to domestic cycles and clinical manifestation such as chagasic cardiomyopathy [35].

Reviewer 3 Report

Comments and Suggestions for Authors

Comments are uploaded. 

Author Response

This is an interesting report on the use of antigens from local Trypanosoma cruzi

parasite strains and molecular tools for the detection of the parasite in blood samples.

The report has several notable strengths including:

  1. The use of local strains in ELISA for accurate patient diagnosis.
  2. The use of PCR to validate the ELISA data.

However, several areas in the text and data analysis require improvements.

Major comments:

  1. The authors have not indicated how many times the ELISA experiments were done. They experiments should be done three times, and the average of the results should be reported with error bars. It is important to determine to validate the “in house” antigens compared to the Chagas III ELISA diagnostic kit.

Answer: Thank you for your thoughtful comment. Due to our laboratory's established standardization protocols for in-house ELISAs, each serum sample was tested individually against each antigen. These protocols ensure both technical repeatability (triplicate wells) and biological repeatability (sample repetition on different days). The commercial assay was optimized for single-well sample testing based on manufacturer's instructions. In addition, in the revised Figure 1, we now present the data as DO/cut ratios, facilitating direct comparison between samples.

  1. There should be a more thorough explanation of the data shown in Figure 3. The

author failed to explain the conclusions from the phylogenetics analysis. What is

  1. cruzi from the patient blood closest to?

We have now added that “indicating that is closely related to a TcI a sequence from a patient from Yucatan, Mexico” (lines 228-229).

Minor Comments:

  1. The title should be changed to “Detection of Trypanosoma cruzi in a reactive blood bank sample in Sonora, Mexico”

Answer: The changes were made accordingly.

Page 1, Lines 2-3…..Detection of Trypanosoma cruzi in a reactive blood bank sample in Sonora, Mexico

  1. In line 22: the word “considered” should be removed.

Answer: The changes were made accordingly.

Page 1, Line 22… Chagas disease is a neglected disease caused by the parasite Trypanosoma cruzi, a public

  1. In line 28: antigenic should be changed to antigens.

Answer: The changes were made accordingly.

Page 1, Line 28… using an “in-house” ELISA, which employs three different antigens total extract

  1. Line 29: the should be a period after Brenner. The word and should be removed, and the sentence stars with “The molecular characterization…”

Answer: The changes were made accordingly.

Page 1, Line 30.…strain H1 and CL-Brener. The molecular characterization of Trypanosoma cruzi…

  1. Line 60: add by between monitor and epidemiological

Answer: The changes were made accordingly.

Page 2, Line 86… and monitor by epidemiological surveillance.

  1. Line 172, should be changed to “The risk of T. cruzi infection after…”

Answer: The changes were made accordingly.

Page 7, Line 258… The risk of T. cruzi infection after a transfusion…

  1. Line 134: change to anti-Trypanosoma cruzi antibodies

Answer: The changes were made accordingly.

Page 5, Line 213….anti-Trypanosoma cruzi IgG antibodies

  1. Line 62: antigenic should be changed to antigen.

Answer: The changes were made accordingly.

Page 3, Line 111… which employs as antigen a total extract from Trypanosoma…

  1. In the Discussion, in Line 175, the authors wrote that 163.416 transfusions were recorded from 2018-2020. Yet, in the Line 176, the authors write that “16000 people could be at risk of acquiring the infection”. Why are 16,000 people at risk?

Answer: We thank the reviewer for identifying the error. The reported value was incorrect due to an unintentional omission. The correct at-risk population is 160,000, and the manuscript has been updated accordingly.

Page 7, Lines 261-262… This suggests that, without proper screening, more than 160,000 people could be at risk of acquiring…

Reviewer 4 Report

Comments and Suggestions for Authors

I have attached my comments in a separate Word document

Author Response

I believe this is an interesting report on a topic of increased interest from the point of view of Tropical Medicine in general, and Chagas disease in particular. I think that with a few minor corrections, it will be ready for publication. For the convenience of authors, I have arraged my comments in specific (line-by-line) and general.

Specific comments

Line 23-24- please rephrase to “…, Chagas disease is considered endemic in the southern region of the country” or something to that effect.

Line 54- please add a reference to justify that statement

Answer: The changes were made accordingly.

Page 2, Lines 74-77… Although blood bank screening has been regulated in Mexico since 2012, according to the Official Mexican Standard (NOM 253-SSA2-2012), few confirmed cases have been reported, even though infected triatomines have been reported in many regions of the country [19].

Line 57- same above

Answer: The changes were made accordingly.

Page 2, Lines 77-80… Today, screening for Chagas is only carried out on blood donors, which leaves most of the population unaware of whether they are infected and, if they are infected, without access to treatment [20].

Line 80- the formula for sulfuric acid is not written/formatted correctly.

Answer: The changes were made accordingly.

Page 3, Line 135…15 min with 1 M sulfuric acid (H2SO4)

Line 86- here, and in other cases in the text you write “qPCR” and refer to real-time PCR”; even thought real-time PCR is the same as qualitative PCR, this must be clarified in the text.

Answer: The changes were made accordingly.

Page 4, Line 145…PCR (qPCR), also known as quantitative PCR,

Line 112- if your case of Chagas disease in the region of note, how can there be already a previous relevant reference? Perhaps ref 8 should be removed earlier in the sentence?

Answer: The changes were made accordingly.

Page 4, Lines 177-178…Although 65 years have passed since the first case of CD was reported in Guaymas, Sonora [29], this study describes for the first time the presence of T. cruzi in a volunteer

Line 114- from your phrasing it is not clear if the patient was unaware of the existence of triatomine vectors in his residence, or if was unaware that these are the vectors of Chagas disease. Please clarify.

Answer: The changes were made accordingly.

Page 4, Lines 180-181… He reported having pets at home but did not recognize nor know triatomine vectors.

Line 154- please explain briefly in the text the meaning of Discrete Typing Unit with a relevant reference. Also, you mind consider abbreviating in DTU after its first mention.

Answer: The changes were made accordingly.

Page 2, Lines 51-54… T. cruzi exhibited great genetic variability, is classified into seven Discrete Typing Units (DTU) from TcI-TcVI and Tc-Bat [5,6]. The term is defined as “sets of stock that are genetically more related to each other than to any other stock and that are identifiable by common genetic, molecular or immunological markers” [7].

General comments

  1. Some references, namely 1 & 3 are not formatted properly, and it also seems to me that they can be replaced by proper papers in scientific journals. Please make the necessary changes.

Answer: The changes were made accordingly. The reference 1 was changed as per suggestion.

Page 8, Lines 302-305… Nepomuceno de Andrade, G.; Bosch-Nicolau, P.; Nascimento, B.R.; Martins-Melo, F.R.; Perel, P.; Geissbühler, Y.; Demacq, C.; Quijano, M.; Mosser, J.F.; Cousin, E.; et al. Prevalence of Chagas disease among Latin American immigrants in non-endemic countries: an updated systematic review and meta-analysis. Lancet Reg Health Eur 2024, 46, 101040, doi:10.1016/j.lanepe.2024.101040.

Answer: The changes were made accordingly. The reference 3 was formatted properly as per suggestion.

Page 9, Lines 353-356.. Secretaría de Salud de México. Incidencia de Tripanosomiasis americana crónica (Enfermedad de Chagas) (B57.2-B57.5) por grupos de edad Estados Unidos Mexicanos 2023. Anuarios de Morbilidad 1984 a 2023. 2023. Disponible en: https://epidemiologia.salud.gob.mx/anuario/2023/incidencia/enfermedad_grupo_edad_entidad_federativa/187.pdf (accessed on February 24, 2025).

  1. Explain briefly what you mean by the term “in-house” ELISA, and how this method can be further modified to fit in the needs of different regions and laboratories.

Answer: The changes were made accordingly.

Page 3, Lines 119-120…The presence of anti-T. cruzi antibodies were analyzed by an “in-house” ELISA, which is a customized assay

  1. I think it would be beneficial to briefly discuss if treatment for Chagas disease, albeit incomplete, would be applicable for other Trypanosoma spp. Infections.

Answer: The changes were made accordingly.

Page 4-5, Lines 188-193… The treatment of Chagas disease has more than 50 years and relies on two drugs: BNZ and NFX. These have great effectiveness in acute infections but significant side effects due to the oxidative stress reactive metabolites generated, which can eliminate T. cruzi, but harm the organism [2]. It is not clear whether these drugs can effectively curet patients infected with other Trypanosoma species due to metabolic differences, but they have been tested in studies as a combination of drugs [30,31].

Round 2

Reviewer 3 Report

Comments and Suggestions for Authors

The authors have adequately addressed the concerns noted in the manuscript.  The manuscript is worthy of publication.